# Diversity of Cytochrome *c* Oxidase Assembly Proteins in Bacteria

**DOI:** 10.3390/microorganisms10050926

**Published:** 2022-04-28

**Authors:** Lars Hederstedt

**Affiliations:** The Microbiology Group, Department of Biology, Lund University, Sölvegatan 35, SE-223 62 Lund, Sweden; lars.hederstedt@biol.lu.se

**Keywords:** cytochrome oxidase, heme protein, copper protein, enzyme biosynthesis, bioenergetics, enzyme assembly factors

## Abstract

Cytochrome *c* oxidase in animals, plants and many aerobic bacteria functions as the terminal enzyme of the respiratory chain where it reduces molecular oxygen to form water in a reaction coupled to energy conservation. The three-subunit core of the enzyme is conserved, whereas several proteins identified to function in the biosynthesis of the common family A1 cytochrome *c* oxidase show diversity in bacteria. Using the model organisms *Bacillus subtilis*, *Corynebacterium glutamicum*, *Paracoccus denitrificans*, and *Rhodobacter sphaeroides*, the present review focuses on proteins for assembly of the heme *a*, heme *a*_3_, Cu_B_, and Cu_A_ metal centers. The known biosynthesis proteins are, in most cases, discovered through the analysis of mutants. All proteins directly involved in cytochrome *c* oxidase assembly have likely not been identified in any organism. Limitations in the use of mutants to identify and functionally analyze biosynthesis proteins are discussed in the review. Comparative biochemistry helps to determine the role of assembly factors. This information can, for example, explain the cause of some human mitochondrion-based diseases and be used to find targets for new antimicrobial drugs. It also provides information regarding the evolution of aerobic bacteria.

## 1. Introduction

The biosynthesis of enzymes containing metal prosthetic groups, such as heme, iron-sulfur clusters, and copper centers, generally requires one or more assisting proteins. Cytochrome *c* oxidases of the heme–copper oxygen reductase family A (present in mitochondria and many aerobic bacteria [1]) is, with respect to the core, a conserved enzyme, as demonstrated by three-dimensional structures, functional features, and the gene sequence information available from a large number of different types of organisms [1,2,3,4,5]. Not long ago, it was assumed that the proteins that function in the biosynthesis of cytochrome *c* oxidase are also conserved [6,7]. However, and intriguingly, there seems to be considerable diversity of some of the assembly proteins and their occurrence in bacteria. This review addresses this diversity and raises questions about how it relates to the evolution of cytochrome *c* oxidase and its specific assembly factors. It also presents an overview of the current knowledge concerning biosynthesis proteins for family A cytochrome *c* oxidase in bacteria.

Research in the field of enzyme biosynthesis can reveal hitherto undiscovered mechanisms in cells and, for example, explain how certain mutations cause disease. From the applied perspective, the diversity of proteins with key functions in the assembly of heme–copper oxygen reductases provide the possibility to find new drugs that specifically affect oxidase function in only certain selected bacterial species and not in eukaryotes.

Cytochrome oxidases in mitochondria and aerobic bacteria function to reduce molecular oxygen to yield water and generate energy conservation by the formation of a transmembrane electrochemical gradient that drives ATP synthesis and other energy-demanding processes in cells. The family A cytochrome *c* oxidases contain three different transmembrane polypeptides, subunits I, II, and III (plus, most often, a small fourth polypeptide), two heme A molecules (heme *a* and heme *a*_3_), and three copper atoms (two in the Cu_A_ center and one in the Cu_B_ center) (Figure 1). Notably, heme A as an enzyme prosthetic group is only found in heme–copper oxygen reductases [8]. Cytochrome *c* oxidase in the mitochondrial respiratory chain (known as Complex IV) contains, in addition to the core subunits, many additional proteins, none of which seem to have a bacterial origin. During respiration, reduced cytochrome *c* donates electrons to cytochrome *c* oxidase via the di-copper center Cu_A_ located in subunit II. From there, the electrons are within the enzyme transferred to subunit I, first to the low-spin heme *a*, then to the dioxygen reduction site containing the high-spin heme *a*_3_ and a copper atom, Cu_B_.

The focus of this review is on proteins that assist in the assembly of the redox metal centers in subunits I and II of subtype A1 cytochrome *c* oxidase in bacteria after the two polypeptides have been inserted into the cytoplasmic membrane. The well-studied bacteria *Bacillus subtilis*, *Corynebacterium glutamicum*, *Paracoccus denitrificans*, and *Rhodobacter (Cereibacter) sphaeroides* are used here as model bacteria to illustrate conservation and diversity of biosynthetic proteins. In evolutionary terms, the taxa of these four model bacteria are late-diverging among members of their *phyla*, and their subunit I polypeptide basically follow such a phylogenetic positioning [9]. Mitochondria probably originated from an α-proteobacterial endosymbiont. The biosynthesis of cytochrome *c* oxidase in the mitochondrion is far more complicated than in bacteria and involves a larger number of proteins of which many are not found in bacteria. One reason for this larger number of biosynthesis proteins is that the genes for subunits I, II, and III in eukaryotes are present in the mitochondrial genome, whereas those for the other subunits and the assembly proteins are encoded in the nuclear DNA and are, after synthesis, translocated from the cytoplasm to the inner mitochondrial membrane. For reviews on mitochondrial cytochrome *c* oxidase biogenesis, see references [10,11,12,13].

## 2. The Model Bacteria *B. subtilis*, *C. glutamicum*, *P. denitrificans*, and *R. sphaeroides*

*B. subtilis* and *C. glutamicum* are Gram-positive bacteria belonging to Bacillota (formerly Firmicutes) and Actinomycetota (formerly Actinobacteria), respectively. *P. denitrificans* and *R. sphaeroides* are Gram-negative α-Proteobacteria. These four model bacteria do not cause disease in animals or plants and contribute to the degradation of organic material in nature. A high G+C content of genomic DNA is a common feature of *C. glutamicum* and *P. denitrificans*. *B. subtilis*, *C. glutamicum*, and *P. denitrificans* are under all conditions organoheterotrophs with mainly a respiratory metabolism [14,15,16]. *R. sphaeroides* is a purple non-sulfur photoheterotroph that, in the dark, is a respiring organoheterotroph [17]. All four bacterial species can also respire anaerobically by nitrate reductase. The cytochrome *aa*_3_ oxidases from *B. subtilis*, *C. glutamicum*, *P. denitrificans*, and *R. sphaeroides* are characterized in the details, including the atomic level structural data [18,19,20,21]. A major experimental advantage of the four model bacteria is that their family A oxidases are not required for growth, which is essential for mutant studies.

*B. subtilis* contains two heme A-containing oxidases: the menaquinol oxidase cytochrome *aa*_3_ and the cytochrome *c* oxidase *caa*_3_. Both enzymes belong to the subtype A1 of heme–copper oxygen reductases. The cytochrome *aa*_3_ has no Cu_A_ center (it was apparently lost during evolution and replaced by a menaquinone binding site [18]). The situation with two variants of the cytochrome *aa*_3_-type in the same bacterial species is useful for experimental identification of proteins specifically required for Cu_A_ assembly and those that function to incorporate heme A or have a role in the assembly of the heme *a*_3_–Cu_B_ center. The electron donating cytochrome *c* in *B. subtilis* cytochrome *caa*_3_ is a domain of subunit II sitting adjacent to the Cu_A_ domain at the C-terminal end of the protein. Cytochrome *caa*_3_, the cytochrome *bc* complex (QcrABC), and a small membrane-anchored cytochrome *c* (CccA or CccB) are in the cytoplasmic membrane found as a supercomplex that can oxidize menaquinol coupled to the reduction of molecular oxygen [22,23].

Cytochrome *aa*_3_ and cytochrome *bc* form a supercomplex also in *C. glutamicum* [24] and other Actinomycetota, and this complex is important for the stability of each constituent enzyme [25,26]. The *cta* and *qcr* genes encoding the supercomplex polypeptides are organized in one operon (Figure 2). The QcrC subunit of the *bc* complex in Actinomycetota has a diheme cytochrome *c* domain that functions as the electron donor to the cytochrome *aa*_3_ in the supercomplex [21,24]. There seems to be no other cytochrome *c* in *C. glutamicum* [26].

*P. denitrificans* and *R. sphaeroides* belong to the order Rhodobacterales and contain, under high oxygen conditions, one mitochondrial-type cytochrome *aa*_3_ and a water-soluble periplasmic cytochrome *c* as the electron donor to the oxidase. The cytochrome *aa*_3_ and cytochrome *bc*_1_ complex form a supercomplex [27,28]. *P. denitrificans* contains in addition a family B heme–copper oxidase, cytochrome *ba*_3_, whereas *R. sphaeroides*, under low oxygen conditions, contains a family C heme–copper oxidase, cytochrome *cbb*_3_. Cytochromes *ba*_3_ and *cbb*_3_ do not contain the Cu_A_ center. Due to the different cofactor compositions and other differences among cytochromes *aa*_3_, *ba*_3_, *caa*_3_, and *cbb*_3_, they are useful for comparative studies to determine roles of heme–copper oxygen reductase biosynthesis proteins.

## 3. Assembly of Cytochrome *c* Oxidase

The experimental data available, to date, strongly suggests that, in the biosynthesis of family A cytochrome *c* oxidase, subunits I and II mature separately before they associate to form the enzymatically active enzyme [11,29,30]. Subunit I normally has 12 transmembrane segments. After this polypeptide has been inserted into the membrane and folded (to some extent), it is hemylated, to create the hemes *a* and *a*_3_, and one copper ion is inserted to form Cu_B_. Subsequently, a covalent bond essential for proton pumping forms between the ε-N of a His ligand to Cu_B_ and C6 of a Tyr residue at the dioxygen-binding site [31,32,33,34]. It is not clear if any accessory factor is required for this polypeptide modification to occur. Subunit II, with two transmembrane segments, incorporates two copper ions in the extracellular domain to create the Cu_A_ center and the subunit then associates with cofactor-loaded subunit I. The subunit I+II complex binds subunit III, which stabilizes the nascent enzyme [35]. The mature oxidase also contains a tightly bound magnesium ion and bound lipid molecules. Depending on the organism, there are additional modifications of subunit II. In the case of *B. subtilis* cytochrome *caa*_3_, subunit II is a lipoprotein with a diacylglyceride moiety at the N-terminal end (modification achieved by the activities of leader sequence peptidase II and lipoprotein diacylglyceride transferase) and a cytochrome *c* domain (with one covalently bound heme) in the C-terminal end of the protein. Thus, the maturation of the subunit II polypeptide in *B. subtilis* comprises three post-translational modifications. In the final assembled cytochrome *c* oxidase, the heme *a* and heme *a*_3_–Cu_B_ dioxygen-binding center are positioned deep within the membrane, only accessible to small molecules. This feature implies that the processes of insertion of the two heme A molecules and folding of the subunit I polypeptide are integrated. The Cu_A_ and cytochrome *c* (when present) domains of subunit II are exposed on the outer side of the cytoplasmic membrane in bacteria to interact with periplasmic proteins and are assembled at this location.

Heme molecules and copper ions are toxic to cells: when in the reduced state and dioxygen is present, reactive oxygen species can be generated. Reduced heme A is more prone for oxidation than heme B due to its higher mid-point redox potential. Chaperone proteins provide protection against the toxicity of heme and metal ions during transport in the cell, from the site of synthesis or acquisition to the final destination in a protein [36]. Heme and copper ion homeostasis in the cell can, in addition, comprise excretion of excess amounts of the metal factors out of the cell. Most aerobic bacteria contain multiple heme proteins. As an example, *B. subtilis* has more than 24 different heme proteins [15]. The mechanisms of heme transport in bacterial cells and how heme is incorporated into proteins to form cytochromes are generally not understood [37,38]. More is known about copper ion trafficking and homeostasis, but the picture is not complete for any bacterium [39,40].

## 4. Difficulties in the Identification of Biosynthesis Proteins

It is often a challenge to identify and study proteins that function in the biosynthesis of enzymes because: (i) these proteins are, in most cases, not associated with the mature enzyme, (ii) assembly processes are transient and mechanistically not apparent in the enzyme product, and (iii) biosynthesis proteins are generally quantitatively minor constituents in cells and their activities problematic to assay. The large majority of the proteins found to function in cytochrome *c* oxidase biosynthesis have been identified through mutations that affect respiration in microbes or cause disease [41,42,43]. A mutant approach to detect the presence and function of proteins is powerful. However, for the design of experiments and the interpretation of data, it is essential to acknowledge the limitations and caveats in that approach. Proteins essential for cell viability and those with a function covered also by another protein are generally missed when relying on single mutant data. Furthermore, a mutation in a biosynthesis protein for an enzyme might cause indirect effects, which can differ from those resulting from mutations in the actual enzyme. For example, a defect in a biosynthesis protein may result in the accumulation of a toxic intermediate or degradation product that does not appear when the enzyme itself is inactivated. In a mutant approach, we, in addition, need to consider the difference between the complete absence of a protein (due to interruption or deletion of the corresponding gene) compared to the inactivation of a protein by a point mutation (causing an amino acid residue substitution). The complete lack of a certain protein in the cell may disturb the assembly or stability of other proteins and thereby affect several processes. Thus, mutations in a gene can readily cause indirect or pleiotropic effects, leading to the wrong interpretation of data and incorrect suggested function for the protein encoded by that gene. Desired amino acid substitutions in the protein of interest are those that affect the specific activity without gross changes in the structure of the protein [44]. Ideally, to be very informative, a mutant protein in the cell should result in a phenotype and retain most of its properties, such as stability, the ability to bind cofactors, and the interaction with other proteins.

**Table 1 microorganisms-10-00926-t001:** Diversity and occurrence of proteins identified to function in cytochrome *c* oxidase biosynthesis in four bacteria. The color background quickly brings an overview on the occurence of the different proteins among the four bacteria. The UniProt identification numbers bring a second layer (more detailed) of information.

Function		Protein	Bacterium and UniProt Identification Numbers			
			*Bacillus subtilis*	*Corynebacterium* *glutamicum*	*Paracoccus* *denitrificans*	*Rhodobacter* *sphaeroides*
Cu_A_ synthesis		Sco	P54178 (YpmQ)		A1BAG3/A1B5S0 (ScoA/ScoB) ^1^	Q3J6C2 (PrrC)
		CtaK	P40768			
		PCuAC		Q8NPY8	A1BAG4/A1AZD7 (PCu1/PCu2)	A0A2W5SGC1
Cu_B_ insertion		Cox11_CtaG			A1BA38	Q3J5F7
		Caa3_CtaG ^2^	O34329	Q8NMV8 (CtiP) ^3^		
Heme A synthesis and insertion	Heme O synthase	CtaB ^4^	P24009	Q8NQ66	A1BA40	Q3J5F9
	Heme A synthase	CtaA	P12946	Q8NQ70	A1B8C2	Q3IXW9
		Surf1		Q8NNG3	A1BA36 (Surf1c) ^5^	Q3J5F5
Other role		CtaM	O31845			
Cytochrome *c* synthesis	Disulfide bond reduction	CcdA	P45706	Q8NT70	A0A533I729	Q3J4J7
		ResA	P35160	Q8NT71		
		CcmG			P52236	Q3J512
		CcmH			A1B950	Q9ANS4
	Heme transport and ligase	System I: CcmA CcmB CcmC CcmD CcmE			P52218 P52219 P52220 P52221 A1B946	O33570 Q3J515 Q3J514 Q3J513 Q3J278
		System II: CcsB/ResB CcsA/ResC	P35161 (ResB) P35162 (ResC)	Q8NT69 Q8NT68		

^1^ The two *P. denitrificans* Sco paralogs seemingly overlap in functions. ^2^ The role of this protein in Cu_B_ assembly is tentative. ^3^ CtiP is a fusion protein, including a Caa3_CtaG domain, and might function in Cu_A_ rather than Cu_B_ assembly [45]. ^4^
*B. subtilis* contains also a CtaB paralog, CtaO [46]. ^5^
*P. denitrificans* contains two Surf1 proteins, Surf1c and Surf1q, which are specific for the biosynthesis of cytochrome *aa*_3_ and *ba*_3_, respectively [47].

Bona fide cytochrome *c* oxidase biosynthesis proteins function directly and specifically in the assembly of the enzyme by interacting with subunit polypeptides to insert cofactors or to chaperone folding or oligomerization of subunits in the membrane. Other proteins, in oxidase assembly, play a more peripheral role by performing tasks, such as catalyzing the synthesis of a cofactor or the delivery of a cofactor in the appropriate redox state to the proper subcellular compartment. To establish the function at the molecular level of a biosynthesis protein, detailed characterizations of the protein and its activity in vitro with purified components are required. This ultimate goal has essentially been reached for some Cu_A_ center assembly proteins [48], but is remote for many other cytochrome *c* oxidase biosynthesis proteins [49].

In a recent genome-wide approach to find genes important for the biosynthesis of family A heme–copper oxygen reductase, a library of 3966 *B. subtilis* strains, systematically deleted for all non-essential genes, was screened for cytochrome *c* oxidase activity [50]. Additionally, two collections of strains containing random point mutations induced by chemical mutagenesis were screened. The screens revealed two new biosynthesis proteins, making a total of twelve proteins presently known to function in cytochrome *caa*_3_ biosynthesis in *B. subtilis* (Table 1). Three of the proteins (Sco, Caa3_CtaG, CtaK) specifically function in the assembly of cytochrome *caa*_3_; two are the enzymes for heme A synthesis (CtaA, CtaB); four are for the cytochrome *c* synthesis (CcdA, ResA, ResB, ResC); one (CtaM) is required for both cytochrome *caa*_3_ and cytochrome *aa*_3_ activity; and two act in lipoprotein modification (LspA, Lgt) [50]. As discussed earlier in this section, there are probably additional biosynthesis genes in *B. subtilis* to be discovered, because the used screens would not identify genes encoding proteins that overlap in function nor genes that are required for growth or not necessary for cytochrome *caa*_3_ assembly under the growth conditions used.

## 5. Heme Synthesis

Aerobic organisms, with only a few exceptions, can synthesize protoheme IX (heme B). In the bacteria, the tetrapyrrole universal precursor 5-aminolevulinic acid is synthesized either from glycine and succinyl-CoA (called the Shemin pathway) or from glutamyl-tRNA (called the C5 pathway). The synthesis of uroporphyrinogen III from 5-aminolevulinic acid is catalyzed by a conserved set of three enzymes [51]. The subsequent biosynthesis of heme B from uroporphyrinogen III occurs via at least three different pathways (i to iii), depending on the organism [52]. (i) Eukaryotes and Pseudomonadota (formerly Proteobacteria) have the so called classical pathway comprising three enzyme-catalyzed steps and where the insertion of one ferrous ion into protoporphyrin IX occurs in the last step. (ii) In Bacillota and Actinomycetota, the iron atom is inserted into uroporphyrin III forming coproheme that, in the last step of the pathway, is oxidized to yield heme B. (iii) In some bacteria, heme B is synthesized via siroheme in the so called alternate pathway. Notably, in the context and perspective of this review, the three pathways for heme-B synthesis are mosaic, in that they rely on some enzymes that are the same for all pathways and some that are unique for the respective pathway [53]. For a discussion on the possible origin and evolution of enzymes for heme synthesis and heme-containing proteins, see [54].

Heme A is synthesized from heme B in two enzyme-catalyzed steps, with heme O as a stable intermediate [55,56]. The first step is catalyzed by heme O synthase (CtaB/Cox10), a membrane protein that farnesylates heme B [57]. Heme O is then converted to heme A by the activity of heme A synthase (CtaA/Cox15) [8]. Both enzymes have multiple transmembrane segments and are conserved among organisms, although there is considerable sequence variation and the two enzyme polypeptides are fused in some bacteria [9]. Furthermore, in certain archaea, the CtaA polypeptide is only about half the length, compared to what is normal, and forms a homodimer [58,59]. Heme A synthase activity seems based on radical chemistry and requires molecular oxygen, but the oxygen atom incorporated to form the characteristic aldehyde group of heme A is derived from water [60]. The X-ray crystal structure of *B. subtilis* CtaA is known [61]. CtaA and CtaB can form a complex in the membrane and this presumably facilitates the tunneling of heme O between the two enzymes. The complex formation with CtaA and CtaB of *B. subtilis* and *R. sphaeroides* indicates the conservation of the protein structure [62]. Possibly, CtaA binds the oxidase apo-subunit I to transfer newly synthesized heme A for the assembly of hemes *a* and *a*_3_ (Figure 1) [47,55,63].

*E. coli* cells can synthesize heme O, but not heme A. The *E. coli cyoABCDE* operon encodes the four polypeptides of cytochrome *bo*_3_ (a heme O-containing family A heme–copper oxidase). The expression of the cloned *E. coli cyoABCD* in *P. denitrificans* results in assembled functional cytochrome *ba*_3_, i.e., oxidase with heme A in the high-spin site, instead of heme O [64]. In *Bacillus* species, heme O can substitute for heme A in oxidases [65,66]. The findings show flexibility in heme bindings sites in subunit I, and indicate that the availability and stoichiometry of different hemes in the cell influence the heme composition of cytochrome oxidase and thereby enzyme activity.

## 6. Insertion of Heme A into Subunit I

Patients with Leigh’s syndrome show decreased cytochrome *c* oxidase activity in the mitochondria and have a gene for heme A synthesis (CtaB/Cox10 or CtaA/Cox15) or the protein Surf1 (from *SURFEIT* locus) mutated [42]. This suggests that Surf1 has a role related to heme A. Surf1 is a membrane protein with a large extracytoplasmic domain flanked by one transmembrane segment on either side [47]. Surf1 is present in bacteria belonging to the *phylum* Actinomycetota, including the species *C. glutamicum*, *Mycobacterium tuberculosis*, and *Streptomyces coelicolor*, and Pseudomonadota, for example, *P. denitrificans* and *R. sphaeroides*, but is not found in bacteria of the *phylum* Bacillota, for example, *B. subtilis* and *Staphylococcus aureus* (Table 1) [47,67,68]. The effect from mutations in the Surf1 gene varies from moderately decreased cytochrome *c* oxidase activity to an entire lack of this activity, depending on the bacterial species.

*P. denitrificans* contains two Surf1 paralogs, Surf1c and Surf1q, whose respective role is specific for the cytochrome *aa*_3_ and the cytochrome *ba*_3_, respectively [69]. The *surf1c* gene is in the chromosome genetically linked to genes for subunits II and III of cytochrome *c* oxidase (Figure 2). The properties and function of the *P. denitrificans* Surf1 proteins have been analyzed in great detail in the laboratory of Ludwig [6,29,69,70]. For a review see [47]. Surf1c and Surfq can both bind heme A, specifically, and conserved residues important for this function are positioned in the regions where the extracellular domain is attached to the transmembrane segments and includes a His residue, which probably functions as an axial ligand to the heme. It has been suggested that *P. denitrificans* Surf1c acts to modulate heme A synthase activity, store heme A transiently, and to chaperone the insertion of heme A into subunit I [29,47]. Research with *C. glutamicum* has demonstrated that Surf1 is required for the assembly of the cytochrome *bc*–*aa*_3_ supercomplex [68]. The deletion of the *surf1* gene in *R. sphaeroides,* in contrast, has a less drastic effect. It results in three variants of cytochrome *aa*_3_ in the cells: 40% are active enzymes with normal composition, 50% lack the heme *a*_3_, and 10–15% contain the heme *a*_3_ but lack Cu_B_ [67].

Thus, the functional role(s) of Surf1 remains unclear, and it is not known how the two heme A groups are inserted into subunit I after synthesis on CtaA, as discussed in recent reviews [38,55]. In the absence of Surf1, sufficient heme A is in certain cases apparently not available for assembly of the cytochrome *c* oxidase, resulting in a total block in assembly of active enzyme or the accumulation of oxidase assembly intermediates. The composition of the intermediates indicates the order of events in the maturation of subunit I: the low-spin heme *a* forms first, followed by heme *a*_3_ and then the insertion of Cu_B_ (this order is not indicated in Figure 1) and association of subunits I and II containing their cofactors.

## 7. Proteins for Assembly of the Cu_A_ Center

Residues in the conserved sequence HX_34_CXEXCX_3_HX_2_M (X is different amino acids) are directly involved in coordinating the Cu_A_ center. The assembly of Cu_A_ can occur in the absence of protein factors, provided that ligating Cys residues in the site are in the thiol state and copper ions are available [71]. The biosynthesis of Cu_A_ has been elucidated in molecular detail by Hennecke, Glockshuber, and their coworkers using *Bradyrhizobium diazoefficiens* and experiments with mutant cells and isolated components [48]. Based on their findings and many more from other laboratories, the assembly of Cu_A_ involves two periplasmic proteins, Sco and PCuAC (or another small copper-binding protein) [48,72].

Sco (from the synthesis of cytochrome oxidase) is a small copper ion-binding protein with a thioredoxin fold and a CXXXCX_n_H sequence motif [73]. The protein is anchored to the membrane by a single transmembrane segment or by a diacylglycerol moiety (bacterial lipoprotein anchor). Sco is present in many aerobic organisms (other names for Sco are YpmQ, PrrC, and SenC) (Table 1) and its gene is sometimes genetically linked to genes for cytochrome *c* oxidase [74]. PCuAC (from the periplasmic Cu_A_ chaperone) family members are small water-soluble periplasmic copper ion-binding proteins, with a cupredoxin-like fold with an inserted solvent-exposed β-hairpin and His and Met residues in a H(M)X_10_MX_21_HXM sequence motif. A C-terminal unstructured extension rich in His and Met residues is often present [75,76]. For the assembly of Cu_A_, Sco and PCuAC cooperate. First Sco, with one bound Cu^2+^, associates with reduced subunit II. The formed complex triggers the binding of PCuAC loaded with one Cu^1+^ (bound to the structured cupredoxin-like domain) and one Cu^2+^ (bound to the C-terminal part of the protein). Next, the Cu^2+^ is transferred from PCuAC to the Sco/Cu^2+^- subunit-II complex and Sco/Cu^2+^ is released. Finally, a second molecule of copper-loaded PCuAC binds the subunit-II/Cu^2+^ intermediate and one Cu^1+^ is delivered, resulting in the mature Cu_A_ center. There are variations in the mechanism depending on the bacterium [48].

*P. denitrificans* contains two Sco-family and two PCuAC-family proteins (ScoA/ScoB and PCu1/PCu2) [77]. The paralogs seem to overlap in function: mutants lacking ScoA or ScoB show no oxidase defect, and the deletion of both Sco proteins (or growth of a *scoB* deletion mutant in medium with a low copper content) results in oxidase deficiency. In *R. sphaeroides*, the predominant role of PCuAC is to deliver copper ions to Sco (PrrC), which has been shown to be important for maturation of Cu_A_ in cytochrome *aa*_3_ and Cu_B_ in cytochrome *cbb*_3_, and to the Cox11_CtaG protein, which is important for Cu_B_ assembly [78]. *C. glutamicum* apparently has no Sco homolog, but the Cg1883 protein is of the PCuAC-type and encoded by the same operon as CopC (Cg1884), and therefore it probably functions in the assembly of the Cu_A_ center of the cytochrome *bc*–*aa*_3_ supercomplex [45].

Sco (YpmQ) was originally discovered in *B. subtilis* [79] and the protein has been studied in molecular detail [73,80,81]. It is a lipoprotein required for cytochrome *c* oxidase assembly, if the copper ion concentration of the growth medium is ≤1 µM [79,82]. *B. subtilis* does not contain a PCuAC homolog (Table 1) and, instead, has the recently discovered lipoprotein CtaK [50]. CtaK contains conserved His and Met residues, putatively involved in copper ion binding, and has, at the C-terminal end, a peculiar sequence (-EEEHSHHH) which also might contribute to copper ion binding. Strains with the *ctaK* gene deleted, or with a point mutation in the gene (His118→Tyr), show the same phenotype as mutants deficient in Sco: they are defective in cytochrome *c* oxidase activity, contain decreased amounts of full-length subunit II polypeptide relative to subunit I (indicating that subunit II is prone to degradation in the mutants), and are complemented by copper ions added to the growth medium. Compared to Sco-deficient mutants, much lower concentrations of copper ions suppress the phenotype of CtaK-deficient mutants [50].

TlpA is a thiol-disulfide oxidoreductase originally identified in *Bradyrhiozobium japonicum* as being required for cytochrome *c* oxidase activity [83]. The protein keeps the Cys residues in the Cu_A_-binding site in apo-subunit II reduced and also reduces Sco [84]. A TlpA homolog is not found in many bacteria, for example, *B. subtilis*. Other disulfide oxidoreductases, therefore, presumably reduce apo-subunit II or the demand for reductant varies, depending on the redox environment on the outer side of the cytoplasmic membrane [48]. Sco might be a bifunctional protein acting both as a reductant for the Cys residues at the Cu_A_-binding site and as a copper ion-binding chaperone [85]. In *B. subtilis*, the extracytoplasmic BdbD (together with BdbC) catalyzes disulfide bond formation in secreted proteins and in membrane protein domains exposed on the outer side of the cytoplasmic membrane [86,87]. The CcdA protein transfers reducing equivalents from thioredoxin (TrxA) in the cytoplasm to the outer side of the cytoplasmic membrane to reduce ResA and StoA [88] and might also reduce Sco. Among the extracytoplasmatic thiol-disulfide oxidoreductases in *B. subtilis* (BdbD, ResA, StoA, and YneN), only ResA is required for assembly of active cytochrome *c* oxidase [89,90]. ResA and CcdA function together in cytochrome *c* synthesis, by keeping the two Cys residues in apo-cytochrome *c* reduced. The absence of either of these two proteins can, however, be suppressed by the inactivation of BdbD (or BdbC) or by adding reductant (but not by adding copper ions) to the growth medium [90,91]. Based on the available data, it cannot be ruled out that reduced ResA can break a disulfide bond between the two Cys residues in the Cu_A_-binding site of apo-subunit II.

## 8. Proteins for the Assembly of the Heme *a*_3_–Cu_B_ Center

The copper ion of the heme *a*_3_–Cu_B_ center is ligated by three His residues in the subunit I polypeptide. Two of these residues are adjacent and the third His is the one covalently linked to a Tyr residue in the mature oxidase. Depending on the bacterium, either of two very different proteins, Cox11_CtaG or Caa3_CtaG, seems required for the assembly of Cu_B_. Cox11_CtaG is only found in α-, β-, and γ-Proteobacteria [74], including *P. denitrificans*, *R. sphaeroides* (Table 1), and in eukaryots [49]. Caa3_CtaG is present in *B. subtilis* [82], *C. glutamicum* (as part of the CtiP protein) [45] and, notably, also in many α-Proteobacteria [9]. In contrast to the feature of Sco-defective mutants, those deficient in Cox11_CtaG or Caa3_CtaG cannot be complemented by the addition of copper ions to the growth medium [82,92].

As a cytochrome *c* oxidase assembly factor, the Cox11_CtaG protein was first discovered by mutations in the *COX11* gene of yeast [93]. Its role in Cu_B_ biogenesis in bacteria was demonstrated by studies with *R. sphaeroides* [94]. The relatively small Cox11_CtaG is a copper-binding protein exposed on the outer side of the cytoplasmic membrane and anchored by a single N-terminal transmembrane segment [95]. Studies with the wild-type *R. sphaeroides* Cox11_CtaG and mutant variants [92], combined with analyses of the yeast COX11p [96] and of other orthologues, show that the protein functions as a dimer and the Cys residues in a conserved CXC sequence bind two copper ions in a Cu_2_S_4_ cluster bridging the two polypeptides in the dimer [95]. In biochemical experiments with *P. denitrificans*, dimeric Cox11_CtaG was found to bind subunit I early in oxidase assembly and to remain bound far into the process, suggesting a chaperone function of Cox11_CtaG for insertion of heme A and the copper ion [29,97].

Caa3_CtaG proteins have multiple transmembrane segments [9]. The *B.subtilis* protein has seven predicted transmembrane segments and the gene (*ctaG*) is the terminal one in the *cta* locus (Figure 2). In *Bacillus halodurans,* the genes for Caa3_ctaG and Sco are adjacent on the chromosome, consistent with roles in biosynthesis of cytochrome *caa*_3_. Conserved His residues and one Asp residue in Caa3_CtaG might function to ligate a metal ion [9]. The deletion of *B. subtilis ctaG,* or certain point mutations in this gene, results in the lack of cytochrome *caa*_3_ activity, but does not affect cytochrome *aa*_3_ activity [50,82]. The inactive cytochrome *c* oxidase contains heme *a* and heme *a*_3_ and the α-band light absorption peak for reduced heme *a* is blue-shifted. Based on these observations, it was suggested that Caa3_CtaG mediates the insertion of Cu_B_ in cytochrome *caa*_3_ [82], but this needs to be confirmed by, for example, molecular analysis of the purified oxidase from a *ctaG*-negative mutant. If Caa3_CtaG is a Cu_B_ assembly factor, the experimental data suggest the existence of another protein for assembly of the Cu_B_ in cytochrome *aa*_3_ or no that no factor is required in that case. No specific assembly proteins are known for the *E. coli* ubiquinol oxidase cytochrome *bo*_3_ (which contains Cu_B_), except for the heme O synthase [98,99].

*C. glutamicum* Caa3_CtaG is the C-terminal part of the CtiP protein, which has 16 predicted transmembrane segments, and the region corresponding to segments 7–9 is a homolog of CopD [45]. The function of the N-terminal part of CtiP (including segments 1–6) is unknown. The lack of CtiP destabilizes the *bc*_1_–*aa*_3_ supercomplex and causes inactive oxidase activity [45]. This effect is very similar to that of Surf1 deficiency in *C. glutamicum* [68]. It has been proposed that the CopD domain of CtiP in cooperation with the putative copper-binding protein CopC (cg1884) imports copper ions [45]. CopC is exposed on the outer side of the cytoplasmic membrane and anchored to the membrane by a C-terminal transmembrane segment. According to this model, the copper ion for Cu_B_ is provided from the cytoplasm. The protein Cg1883 is of the PCuAC-type and encoded by the same operon as CopC. It probably functions in the assembly of the Cu_A_ center on the outer side of the cytoplasmic membrane [45]. Thus, CtiP, with its Caa3_CtaG domain, might function in Cu_A_ synthesis and is not (or also) a Cu_B_ assembly factor [45].

The structure of subunit I of cytochrome *cbb*_3_ (family C heme–copper oxidase) is similar to that of family A cytochrome oxidase [100]. Subunit II does not contain Cu_A_. In some bacteria, cytochrome *cbb*_3_ biosynthesis depends on Sco and PCuAC, which may indicate that these periplasmic proteins can function in Cu_B_ assembly [39,101] and that the copper ion for Cu_B_ assembly is obtained from the outer side of the cytoplasmic membrane. In other bacteria, cytochrome *cbb*_3_ synthesis, in contrast to cytochrome *aa*_3_ synthesis, is not dependent on Sco [102]. Whether Sco and PCuAC-type proteins have a direct role in Cu_B_ synthesis in some bacteria remains an open question, as discussed in [101].

## 9. Proteins with an Essential Unassigned Function in the Biosynthesis of Cytochrome *c* Oxidase in Bacteria

The CtaM protein is required for cytochrome *aa_3_* activity in *Staphylococcus aureus* [103] and was recently shown in *B. subtilis* to be essential for the biosynthesis of both cytochrome *aa*_3_ and *caa*_3_ [50]. CtaM is a DUF420 domain integral membrane protein with four or five transmembrane segments, depending on the bacterium. Genes for the protein are not widely distributed in bacteria (not found in *C. glutamicum*, *P. denitrificans* and *R. sphaeroides*) and their presence is correlated with genes for cytochrome oxidase and hemeA synthesis [103]. The *ctaM* gene in *S. aureus* is adjacent to *ctaB*, and in *Bacillus firmus*, it is the terminal gene in the *ctaCDEFM* operon. In *B. subtilis*, however, the gene is not co-localized with a gene relevant for heme–copper enzymes.

Heme *a* and heme *a*_3_ form in subunit I in the absence of CtaM in *B. subtilis*. Notably, a redox-active chromophore, with the absorption maximum at about 585 nm when reduced and of unknown identity, occurs in membranes from mutants deleted for the *ctaM* gene [50]. The chromophore might correspond to heme A bound to CtaA or some other protein. The experimental findings indicate that the CtaM protein is important for the assembly of Cu_B_ or for some other process that occurs late in the assembly of family A heme–copper oxidases [50].

## 10. Cytochrome *c* Synthesis

In *c*-type cytochromes, heme B is covalently linked to protein by two thioether bonds (in a few cases by only one bond) formed by the reaction of the heme vinyl groups, with the Cys residues in a CXXCH sequence motif in the apo-cytochrome polypeptide. The stereo-specific ligation occurs in bacteria on the outer side of the cytoplasmic membrane and requires cytochrome *c* synthase activity, heme, and that the Cys residues are in the reduced state. Extracytoplasmic thiol-disulfide oxidoreductases maintain apo-cytochrome reduced.

Significant for the context of this review, there are at least four different machineries (Systems I–IV) for cytochrome *c* synthesis, and the system in operation depends on the organism and also on the cytochrome *c* variant. For reviews see [104,105,106,107,108,109,110,111]. System I comprises several membrane-bound Ccm proteins and accessory thiol-disulfide oxidoreductases. This system is present in, for example, α- and γ-Proteobacteria. System II is composed of two integral membrane proteins, CcsB/ResB and CcsA/ResC (the two polypeptides are fused in some bacteria), and a module of proteins (CcdA, ResA/CcsX) to keep Cys residues reduced. System II is present in, for example, bacteria belonging to the Bacillota (e.g., *B. subtilis*), cyanobacteria, and δ-Proteobacteria (e.g., *Wolinella succinogenes*). Yeasts, such as *Saccharomyces cerevisiae*, contain in the mitochondrial inter-membrane compartment two homologous cytochrome *c* synthases (System III) specific for the water-soluble cytochrome *c* and the membrane-bound cytochrome *c* of the cytochrome *bc*_1_ complex, respectively. The covalent binding of heme to a Cys residue in the cytochrome *b* subunit of the cytochrome *b*_6_*f* complex of chloroplasts depends on System IV. For certain protists that contain cytochrome *c* genes, the assembly proteins have not been identified. Thus, at least five different machineries apparently operate in nature to synthesize cytochrome *c*. System III is thought to have evolved during or after the mitochondrion originated, as the result of endosymbiosis with an α-Proteobacterium (that contained System I) [110,111]. The CcmC and CcmF proteins of System I and ResC/CcsA of System II might have a common evolutionary origin [112]. The low availability of heme gives preference for System I over System II, but how the two systems have evolved and been distributed among bacteria is unexplained [110].

## 11. How Diverse Are the Machineries for the Assembly of Cytochrome *c* Oxidase Redox Metal Centers?

To summarize the available findings concerning assembly of the redox metal centers in family A1 cytochrome *c* oxidase in bacteria, proteins for heme A synthesis (CtaA and CtaB) are conserved and present in the bacteria that have heme A-containing oxidase. Provided that the environmental copper ion concentration is low Sco and a PCuAC family, or a CtaK-type protein, is generally required for the assembly of the Cu_A_ center in subunit II. Proteins that function in the insertion of metal centers in subunit I or in other aspects of oxidase biosynthesis, however, are poorly understood and appear diverse (Table 1). The Surf1 protein promotes heme A trafficking/insertion and is found in α-Proteobacteria, eukaryotes, and Actinomycetota, but is not found in Bacillota. Cox11_CtaG, important for the assembly of Cu_B_, is present in α-Proteobacteria and eukaryotes, but is not found among Bacillota nor Actinomycetota. The Caa3_CtaG protein, present in Bacillota and Actinomycetota, but not in α-Proteobacteria, is required for the assembly of active cytochrome *c* oxidase and is only tentatively identified as a Cu_B_ assembly factor. The CtaM protein, found in Bacillota, is required for heme A-containing family A1 heme–copper oxygen reductases, but not necessary for the insertion of heme *a* and heme *a*_3_. Additional proteins in bacteria are expected to function in the assembly process, but have not yet been identified due to various experimental limitations.

The observed diversity of cytochrome *c* oxidase biosynthesis proteins among bacteria might reflect the existence of different biosynthetic machineries built upon different proteins, and perhaps of variable complexity, which complete the same task. Alternatively, evolutionary unrelated proteins essentially play the same role (for example, PCuAC and CtaK). A confusing factor can be that similar proteins function in different processes (for example, the Sco-family proteins are important for Cu_A_ synthesis in some bacteria and for Cu_B_ synthesis in other bacteria [72]). Another difficulty for the interpretation of experimental findings is the existence of gaps in our knowledge about cell biochemistry, and that the biosynthesis of oxidases is intertwined with heme and copper homeostasis of the cell. Cytochrome *c* oxidase biosynthesis in bacteria reasonably relies on proteins with broad functional roles in the cell (e.g., in membrane protein maturation and in copper and heme metabolism), in addition to those that are oxidase-specific assembly factors. Proteins with an essential role in the cell can potentially be studied using a biochemical approach, based on in vitro protein synthesis procedures or conditional mutants (e.g., temperature-sensitive mutants or cells in which the concentration of the protein of interest can be drastically increased or decreased during growth).

Thus, the question remains open whether there exist different machineries in nature for the assembly of some of the redox metal centers in family A cytochrome *c* oxidase. Precedence for diversity of machineries is the different systems for cytochrome *c* biosynthesis (Section 10) and by the variation in the pathways for heme B biosynthesis (Section 5).

## 12. On the Evolutionary Origin of Family A Heme–Copper Oxidases and Their Assembly Factors

When, how, and where did cytochrome *c* oxidase originate? Judged from the amino acid sequence, structure, and the prosthetic groups, the core of family A cytochrome *c* oxidase is conserved. The family A enzymes have a low affinity for dioxygen, compared to other respiratory oxidases [113], and originated more than 2 billion years ago, i.e., accompanying the drastic increase in atmospheric oxygen content, a revolutionary change for primordial earth, which occurred about 2.3 billon years (Ga) ago [114,115]. The accompanying consequential decrease in the availability of soluble copper ions put new demands on cell metabolism and probably challenged copper-containing enzyme biosynthesis. The ancestral form of family A heme–copper oxidase has diverged in various phylogenetic branches of life and, apparently, the oxidase-encoding genes have been distributed within and across kingdoms by multiple horizontal gene transfers. Because of the ancient origin and the extensive horizontal gene transfers in the past, it is hard to trace the evolutionary origin of genes for cytochrome *c* oxidase and the topic is unsettled with different interpretations of data prevailing, see [116] and the references therein. Homologs of the cytochrome *c* oxidase biosynthesis proteins found in α-Proteobacteria (CtaA, CtaB, Cox11_CtaG, Sco, and Surf1) are present in eukaryotes, which is consistent with the origin of mitochondria from an ancestor of such a bacterium [49].

The demand for proteins for biosynthesis in a bacterium is arguably influenced by the structure and biochemistry of the cell, combined with the prevailing cell environment. Gram-negative bacteria have a membrane-enclosed periplasmic compartment with a relatively constant environment and protein composition, compared to the corresponding outer surface of the cytoplasmic membrane in Gram-positive bacteria, which essentially is exposed to the bulk of the cell external environment. This difference is reflected in features of various proteins that function on the outer side of the cytoplasmic membrane. For example, the *c*-type cytochromes and the extracytoplasmic thiol-disulfide oxidoreductases are membrane-anchored in Gram-positive bacteria, but mostly water-soluble proteins in the periplasm in Gram-negative bacteria. In view of the four model bacteria in this review, the presence of some (but not all) oxidase biosynthesis proteins is correlated with whether the bacterium is Gram-positive or Gram–negative (Table 1). Notably, as mentioned earlier, no specific proteins have been identified for the assembly of *E. coli* cytochrome *bo*_3_ (a family A heme–copper oxygen reductase). This may suggest that no enzyme-specific factors are absolutely required for assembly of the heme centers and Cu_B_, and that *E. coli*, during evolution, lost factors that were present in the ancestor.

Heme A synthesis must have been present when the heme A-containing oxidase first appeared. The oxidase probably originated from an oxidase with heme O as the prosthetic group(s) similar to cytochrome *bo*_3_. Phylogenetic analysis suggests that subunit I, CtaA, and Caa3_CtaG have coevolved and that heme A-containing cytochrome *c* oxidase originated in an ancestor of extant iron-oxidizing Pseudomonadota [9]. The origin and evolution of genes for assembly factors reasonably depends on whether the biosynthesis proteins are absolutely required. It is conceivable that some initially not essential biosynthesis proteins arouse later, from the need to assemble oxidase more accurately (for example, to avoid the incorporation of heme O or heme B instead of heme A) and at rates compatible with the physiology of the organism and to match copper and heme availability. The acquisition of a new biosynthesis protein can be accomplished by gene duplication (of an entire gene or part of a gene), followed by adaptive evolution of one copy to encode the biosynthesis factor. The evolutionary adaptation of the function of the protein is generally a slow process that occurs before and/or after horizontal gene transfer of the corresponding gene between organisms. In bacteria, the genes for some of the identified biosynthesis proteins are frequently found clustered with those for polypeptides of the oxidase, for example, Sco, Surf1, CtaA, CtaB, Cox11_CtaG, and Caa3_CtaG [74,116] (Figure 2). A clustering of genes promotes the likelihood of their co-transfer and its presence in genomes can reflect past horizontal gene-transfer events, but can also have other explanations. The clustering of genes encoding proteins with associated functions might be beneficial for the co-regulation of transcription and translation of the genes, which is important for the assembly of complex enzymes, such as cytochrome *c* oxidase.

In conclusion, to date, the in bacteria identified heme A-containing family A1 cytochrome *c* oxidase biosynthesis proteins show an apparent mosaic distribution pattern: some proteins are present in most cases, others have a limited distribution and are found in different combinations (Table 1). The mosaic pattern presumably reflects phylogeny as well as variable demands for specific assembly factors and, perhaps, the existence of different processes or types of machineries for the assembly of the same type of redox metal center. Some biosynthesis factors have seemingly coevolved with subunit I of heme A-containing cytochrome *c* oxidase [9], but the evolutionary origins and the dispersion and loss in the past of the factors among bacteria are enigmas.

## Figures and Tables

**Figure 1 microorganisms-10-00926-f001:**
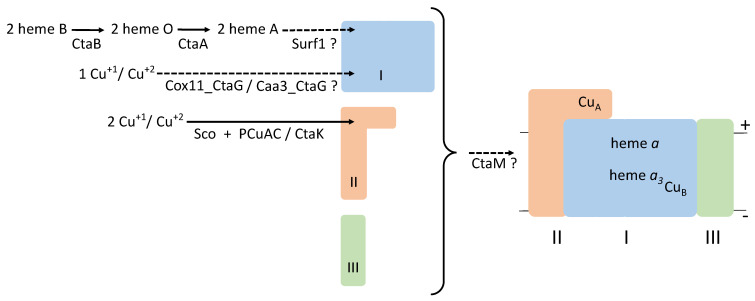
Scheme of cytochrome *c* oxidase biosynthesis in bacteria and the roles of proteins involved in assembly of the oxidase. The mature enzyme is illustrated on the right-hand side with the core three oxidase subunits labeled with roman numerals. Dashed arrows indicate assembly steps that are not understood at the mechanistic level. A question mark indicates that the function of the protein in the assembly of the oxidase is not established. The biosynthesis proteins presented in the figure are not all found in a single bacterium. + and – indicate the outer and inner sides of the cytoplasmic membrane.

**Figure 2 microorganisms-10-00926-f002:**
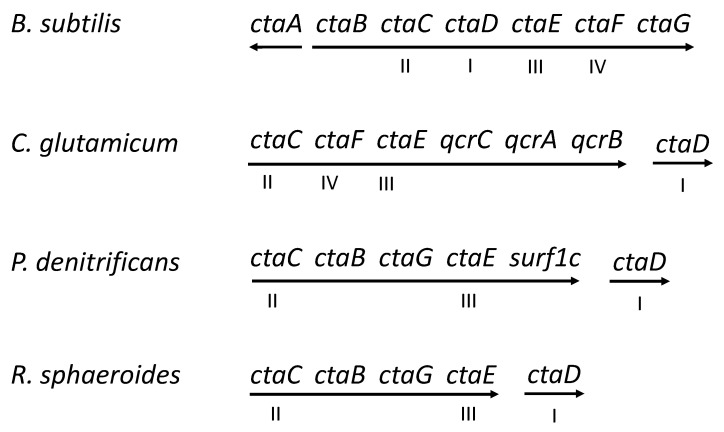
Organization of genes for family A1 cytochrome *c* oxidase in the chromosome of four model bacteria. Arrows show the transcription units. The genes for oxidase polypeptides are indicated I, II, III, and IV. The *qcr* genes encode the polypeptides of the cytochrome *bc* complex. The *ctaG* genes of *B. subtilis* and *P. denitrificans* encode different types of proteins, i.e., Caa3_CtaG and Cox11_CtaG, respectively (see text).

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
