# Peer review of "Diversity of Cytochrome c Oxidase Assembly Proteins in Bacteria"

_microorganisms, 2022, doi:10.3390/microorganisms10050926_

Round 1

Reviewer 1 Report

Congratulations! It is an interesting paper regarding to the model organisms.

I would suggest and ask if You could add some comments according to the following issues:

  1. You have figured out smthg regarding the subunit composition could You add comments about different ways of electron transmission from your research work or other reports?
  2. I would appreciate if You could say smthg about different enzymatic activities of CytOx.
  3. Could You give us some remarks about phosphorylations of the different enzymes and the action of Kinases/Phasphatases during the assembly?

Thanks again for the intresting paper.

Author Response

Response to the specific comments of Reviewer 1:

  1. I do not understand the comment. What is meant by “electron transmission from your research work or other reports”?
  2. The family A cytochrome c oxidases function to oxidize cytochrome c and reduce molecular oxygen. Some oxidases (of other families) can use, for example, NO but these enzymes are not the topic of this review.
  3. As far as I know there is no information available on putative phosphorylation/dephosphorylation of family A cytochrome c oxidase or its biosynthesis proteins in bacteria.

In the revised manuscript the English language and mistakes in the text have been corrected - changes are indicated in the submitted docx file. A small error in Figure 2 has also been corrected.

Reviewer 2 Report

The review integrates in a comprehensive and sound way the knowledge about the proteins involved in the assembly of cytochrome c oxidase. The manuscript is very well written, follows a logical structure, and is supported by many recent references.  my main criticism is that the manuscript can be improved from a visual point of view if two or three figures of some of the pathways mentioned in the paper are added. For example, the synthesis of the heme group, or the 4 systems for the synthesis of cytochrome c. However, I think the manuscript in its present form is ready for publication.

Author Response

Response to the specific comments of Reviewer 2:

Excellent figures on the different pathways of heme synthesis and systems for cytochrome c synthesis are presented in the easy accessible key references 51 and 52, and in references 107, 108 and 110, respectively. I find it superfluous to add such figures to the review.

In the revised manuscript the English language and mistakes in the text have been corrected - changes are indicated in the submitted docx file. A small error in Figure 2 has also been corrected.